# CD38 in Neurodegeneration and Neuroinflammation

**DOI:** 10.3390/cells9020471

**Published:** 2020-02-18

**Authors:** Serge Guerreiro, Anne-Laure Privat, Laurence Bressac, Damien Toulorge

**Affiliations:** Encefa, 16 Avenue des Arts, 94100 Saint-Maur-des-Fosses, France; serge.guerreiro@encefa.com (S.G.); annelaure.privat@encefa.com (A.-L.P.); laurence.bressac@encefa.com (L.B.)

**Keywords:** CD38, NAD, neurodegeneration, neuroinflammation, aging, Alzheimer’s disease, Parkinson’s disease, ALS

## Abstract

Neurodegenerative diseases are characterized by neuronal degeneration as well as neuroinflammation. While CD38 is strongly expressed in brain cells including neurons, astrocytes as well as microglial cells, the role played by CD38 in neurodegeneration and neuroinflammation remains elusive. Yet, CD38 expression increases as a consequence of aging which is otherwise the primary risk associated with neurodegenerative diseases, and several experimental data demonstrated that CD38 knockout mice are protected from neurodegenerative and neuroinflammatory insults. Moreover, nicotinamide adenine dinucleotide, whose levels are tightly controlled by CD38, is a recognized and potent neuroprotective agent, and NAD supplementation was found to be beneficial against neurodegenerative diseases. The aims of this review are to summarize the physiological role played by CD38 in the brain, present the arguments indicating the involvement of CD38 in neurodegeneration and neuroinflammation, and to discuss these observations in light of CD38 complex biology.

## 1. Introduction

Neurodegenerative diseases (NDDs) refer to conditions in which neurons in the central or peripheral nervous system degenerate. Depending on the localization and the neuronal population that degenerates, different pathologies arise, leading to debilitating symptoms and ultimately patient death. While the process by which neurons undergo degeneration remains elusive, it appears to be multifactorial at the cellular level. Indeed, shared cell-autonomous mechanisms were shown to be involved in the degenerative process among NDDs including oxidative stress, excitotoxicity, mitochondrial dysfunctions, as well as autophagy impairment [1]. Furthermore, non-cell-autonomous processes like neuroinflammation are also heavily suspected to participate in neurodegeneration [2]. Due to the lack of effective treatment allowing to stop or at least slow down the neurodegenerative process, NDDs are still an unmet medical need and represent a burden for patients, their relatives, and the healthcare system, resulting in a tremendous economic and societal cost. Most high-profile clinical trials for NDDs led to inefficacious results [3], suggesting that novel approaches to treat these pathologies are needed.

Targeting NDDs through the prism of aging is one of such approach. Indeed, the primary risk factor associated with NDDs, including Alzheimer’s disease, Parkinson’s disease, amyotrophic lateral sclerosis, or Huntington’s disease is aging [4]. Consequently, it is tempting to study age-related dysfunctions that could favor or be instrumental in the neurodegenerative process. Reduced nicotinamide adenine dinucleotide (NAD) levels might be one of these age-related dysfunctions influencing neurodegeneration [5]. Indeed, NAD levels were found to decrease as a consequence of aging [6], including in the human brain and cerebrospinal fluid (CSF) [7,8], while NAD was found to be a potent neuroprotective and anti-inflammatory molecule [9]. The reason as to why NAD levels are reduced as a consequence of aging remained elusive until the discovery in 2016 that expression of CD38, the main enzyme responsible for NAD degradation [10], increased as a consequence of aging, thus explaining age-related NAD decline [11]. Moreover, CD38 deletion was found to repress neurodegeneration and neuroinflammation in experimental models of NDDs. However, CD38 biology is complex and not restricted to its NAD-degrading ability [12]. The aims of this review are to summarize the physiological role played by CD38 in the brain, to discuss whether CD38 is involved in neurodegeneration and neuroinflammation, and to present how to interpret experimental data in view of CD38 complex biology.

## 2. The Complex Biology of CD38

CD38 is a 45 kDa transmembrane glycoprotein composed of a short cytoplasmic tail (amino acid (aa) 1–21), a transmembrane domain (aa 22–42) and an extracellular domain (aa 43–300) [12]. CD38 has both a receptor- and an enzyme-mediated function [12]. As an ectoenzyme, CD38 is a multifunctional protein that catalyzes several reactions: (i) the conversion of NAD into adenosine diphosphate-ribose (ADPR); (ii) the conversion of NAD into cyclic ADPR (cADPR, cyclase activity); (iii) the hydrolysis of cADPR into ADPR; (iv) in the presence of nicotinic acid (NA) and in acidic conditions, the conversion of NADP, the phosphorylated equivalent of NAD, into nicotinic acid adenine dinucleotide phosphate (NAADP); (v) the conversion of NAADP into ADPR phosphate (ADPRP). CD38 is also able to catalyze the degradation of the NAD precursor nicotinamide mono-nucleotide (NMN) into nicotinamide [13]. Of interest, CD38 enzymatic function is pH-dependent [14]. As a receptor, CD38 interacts with its ligand CD31 [15,16]. CD31, also known as platelet endothelial cell adhesion molecule-1 (PECAM-1), is a 130 kDa type I transmembrane glycoprotein that consists of six extracellular immunoglobulin-like homology domains, a 19-residue transmembrane domain, and a 118-residue cytoplasmic tail [17]. CD31 expression is mainly observed in endothelial cells, where it is considered as a constitutive marker [18].

CD38 can be internalized and redirected to the lysosome as well, where the acidic environment shifts CD38 enzymatic activity toward NAADP synthesis [19]. The pathway by which CD38 is internalized might be ligand-dependent since anti-CD38 nanobodies internalize CD38 through a clathrin-dependent mechanism [19] while NAD or thiol compounds act through a non-clathrin-dependent pathway [20]. Whether this internalization can be triggered by CD31 binding or not is suspected but not clearly demonstrated.

## 3. Physiological Role of CD38 in the Brain

### 3.1. CD38 Expression in the Brain

CD38 is expressed in the brain across species including mouse [21], rat [22,23] and human [24]. In the human brain, it is interesting to note that CD38 is expressed in virtually all brain areas, and it is found at statistically significantly higher than average levels in the caudate, pallidum, olfactory bulb, putamen, thalamus, and cingulate anterior [25]. At the cellular level, CD38 is expressed in neurons [22,24], astrocytes [22,26,27,28,29], and microglial cells [30,31,32]. In neurons, CD38 was found mainly in neuronal perikaria but also in dendrites [22,24]. At the subcellular level, in the mouse brain, CD38 is mostly located at the plasma membrane but also present intracellularly [21,33].

During development, CD38 is highly expressed in the mouse brain between postnatal days 14 and 28, mainly in astrocytes, where it regulates their development in a cell-autonomous manner and promotes the differentiation of oligodendrocytes in a non-cell-autonomous manner [34]. CD38 is important for neuronal development as well since abnormalities in the number of neurons and neuronal morphology in the visual cortex and dentate gyrus were reported in CD38 knockout (KO) mice [35].

### 3.2. Physiological Role of CD38 Enzyme Function in the Brain

The CD38 knockout mouse brain displayed a 10-fold increase in NAD levels compared to wild-type mice [33] suggesting that CD38 is one of the main regulators of intracellular NAD levels in the brain. Moreover, using rat cortical neuron cultures, Braidy et al. found that silencing CD38 expression using siRNA increased NAD levels by 5-fold, an effect that was accompanied by a 5-fold increase in the activity of the NAD-dependent enzyme SIRT1 [23]. Thus, through its NADase activity, CD38 controls NAD bioavailability and the activity of NAD-dependent enzymes.

Through its cyclase activity, CD38 produces cADPR, a universal calcium mobilizer involved in neurotransmitter release by neurons and astrocytes [36]. In CD38 KO mice, cADPR synthesis is totally abrogated in the heart, liver, kidney, spleen, or uterus but not in the brain where cADPR levels remained unaffected [33,37,38]. This suggests that CD38 does not act as a cADPR-producing enzyme in the brain. However, this is not verified in all brain areas, since Jin et al. found reduced cADPR levels in the hypothalamus and posterior pituitary of CD38 KO mice. This reduction in cADPR levels in these brain areas led to a reduction of oxytocin and vasopressin release, explaining the observed defects in maternal nurturing and social behavior characteristics of CD38 KO mice [39]. This effect is not limited to the posterior pituitary hormone family of neurotransmitters since norepinephrine, serotonin and the dopamine metabolites dopac and homovanillic acid are strongly increased in prefrontal cortex from CD38 KO mice [40]. CD38 KO mice also had reduced dopamine release in the nucleus accumbens [41]. Altogether, these data indicate that CD38 controls neurotransmitter release through its cADPR-producing activity.

### 3.3. Physiological Role of CD38 Receptor Function

As described by Fang et al., CD38 receptor function influences CD38 enzyme activity because of its pH-dependency, through CD38 internalization and acidic endolysosomal compartment relocation [19]. However, whether or not this phenomenon is at play in brain cells is unknown, likewise whether or not the CD38 ligand CD31 can mediate CD38 internalization.

CD31 expression in the brain is restricted to vascular endothelial cells [42] where it plays a key role in the establishment and the maintenance of intercellular junction between endothelial cells through homophilic binding, as well as in the regulation of vascular permeability [43]. Direct interaction between CD31 on endothelial cells and CD38 on brain cells surrounding cerebral vasculature (pericytes) could have a role in the establishment and/or the maintenance of the neurovascular unit, but no data currently support such an assumption.

CD31 can be cleaved by metalloproteinases, released as a soluble fragment (sCD31) [44], and high sCD31 levels are detected in human plasma samples [45]. Due to the presence of the blood–brain barrier (BBB), sCD31–CD38 interaction in brain cells seems highly unlikely. Yet, since (i) sCD31 is detected in human CSF [46] and (ii) CD38 is mainly located at the plasma membrane level in brain cells [21,33], sCD31 could interact with CD38 located at the plasma membrane of brain cells to trigger CD38 internalization. The physiological function of this putative sCD31/CD38 interaction in brain cells remains elusive. It is worth noting that in pathological conditions in which BBB breakdown is observed like after ischemic stroke, sCD31 levels strongly increased in the brain and were found to correlate positively with (i) neurological stroke severity and (ii) the degree of functional disability [46,47]. From this standpoint, sCD31–CD38 interaction might be harmful to neurons following BBB breakdown, but clear data are lacking. Accordingly, it is interesting to note that plasma sCD31 levels are increased in several neuroinflammatory diseases including multiple sclerosis and HIV-encephalitis [18], but whether such an increase is also observed in CSF remains unknown.

### 3.4. CD38 Role in Neuroinflammation

CD38 expression was found to increase after neuroinflammatory insults. In humans, increased CD38 expression was observed in astrocytes of brain from patients suffering from HIV-1 encephalitis [28]. In vitro, both the proinflammatory cytokine interleukin-1β as well as the HIV-1 glycoprotein 120 increased CD38 expression and enzymatic activity in a dose-dependent manner in human astrocytes [48]. CD38 knockdown using specific siRNAs significantly reduced astrocyte proinflammatory cytokines and chemokines production [28], suggesting that CD38 acts as a regulator of neuroinflammatory processes. Moreover, hydrogen peroxide (H_2_O_2_) treatment also increased CD38 expression in astrocytes, and CD38 knockdown significantly potentiated H_2_O_2_-induced astrocyte death, thus linking oxidative stress, cell survival, and neuroinflammation [26]. CD38 is also involved in the transfer of mitochondria from astrocytes to neurons after stroke, thus linking CD38 to astrocyte-induced neuroprotection [49]. It is worth noting that CD38 KO or inhibition of its enzymatic activity attenuated glioma progression [50,51].

The role played by CD38 in microglial cells is more ambiguous. Like in astrocytes, CD38 expression and enzymatic activity was increased in primary microglial cells following lipopolysaccharide (LPS) and interferon-γ application. On the other hand, CD38 deletion reduced activation-induced microglial cell death [32] and controlled the basal survival of microglial cells [30]. More recently, CD38 knockdown was found to increase apoptosis in normal microglia, but played a protective role in LPS-stimulated microglia and reduced proinflammatory cytokine secretion (Figure 1) [52]. Altogether, these data demonstrate that CD38 plays a central role in microglial cells by linking activation status to cell survival. Whether neuroinflammation modifies CD38 trafficking or internalization remains unknown.

## 4. CD38 Involvement in Neurodegeneration and Neuroinflammation: Evidences

There is no direct genetic evidence linking CD38 to NDDs, and CD38 levels in CSF samples from healthy and NDDs patients has never been assessed. However, reduced NAD levels are a common observation among NDDs including Alzheimer’s disease [53], Parkinson’s disease [54], amyotrophic lateral sclerosis [55], as well as multiple sclerosis [56]. If we consider NAD levels as an inversely correlated marker of CD38 expression and activity [11], this indirectly demonstrates increased CD38 expression in NDDs. Since NAD is a potent neuroprotective and anti-inflammatory agent [5], this constitutes a first indirect proof of CD38 involvement in neurodegeneration and neuroinflammation. However, several other enzymes and pathways regulate NAD levels, which may also be important in NDDs. Consequently, CD38 could be one of several mechanisms involved in the low NAD levels observed in NDDs.

In Alzheimer’s disease, CD38 immunoreactivity was observed in intracellular tangles and neuropil threads [57]. Second, the CSF levels of mi-RNA-708 and miRNA-140-3p, which were shown to reduce CD38 expression in vitro [58,59], are significantly decreased by nearly 4- and 2.5-fold, respectively, in Alzheimer’s disease patients compared to age-matched controls [60]. In Parkinson’s disease, CD157 polymorphism, a paralog gene of CD38, was identified by GWAS studies as a risk factor associated with the disease [61].

Several experimental data using CD38 KO mice demonstrated positive effects of CD38 deletion against neurodegeneration and neuroinflammation. In stroke models induced by focal cerebral ischemia and reperfusion, CD38-deficient mice showed decreased local expression of the proinflammatory chemokine MCP-1, reduced populations of infiltrating macrophages and lymphocytes in the ischemic hemisphere, and, more importantly, reduced cerebral ischemic injury and neurological deficit after three days of reperfusion [62]. Using another model of transient forebrain ischemia, Long et al. [63] observed a significant amelioration in both histological and neurologic outcome following ischemic insult in CD38 KO mice, an effect that could be mediated by increased hippocampal NAD levels detected during reperfusion. By crossing the classical Alzheimer’s disease mouse model APPswePS1ΔE9 with CD38 KO mouse, Blacher et al. [64] found that CD38 deletion reduced Aβ plaque load and soluble Aβ levels, an effect that correlated with improved spatial learning in vivo and that was mimicked by inhibitors that blocked CD38 enzyme activity in vitro. This effect seems paradoxical since CD38 KO mice showed deficits in various learning and memory tasks such as the Morris water maze, contextual fear conditioning, and the object recognition test [65]. In the experimental autoimmune encephalomyelitis mouse model of multiple sclerosis, CD38 deletion reduced disease severity [66]. Note that CD38 deletion was also shown to suppress glial activation and neuroinflammation in a mouse model of demyelination induced by cuprizone administration, an effect that was most likely due to enhanced level of NAD in CD38 KO mice [67].

Altogether, it is interesting to observe that in all these studies, CD38 deletion strongly reduced neuroinflammation, suggesting that CD38 may play a key role in this process. The vast majority of these studies suggest that the beneficial effects observed in CD38 KO mice are explained by increased NAD levels. However, because of the poor characterization of the effect of CD38 deletion in the brain, these results require to be tempered. Indeed, the impact of CD38 deletion on brain cADPR, ADPR, NAADP, and ADPRP levels is not solved and could participate in the beneficial effects observed in CD38 KO mice.

## 5. Aging, the Missing Link between CD38 and Neurodegeneration/Inflammation?

As previously mentioned, there is no direct genetic evidence linking CD38 to NDDs and, unfortunately, no study directly demonstrated increased expression of CD38 in human brain as a consequence of aging or whether CD38 levels are increased in brain from NDDs patients compared with age-matched controls. However, aging is the primary risk factor associated with the vast majority of NDDs [4], and several data tend to indicate that CD38 expression increases in the brain as a consequence of aging. While Camacho-Pereira et al. demonstrated that CD38 expression increases in several tissues as a consequence of aging, the brain was not included in this study [11]. Braidy et al. demonstrated that NADase activity increased in rat brain as a consequence of aging, including in the hippocampus, cortex, cerebellum, and brainstem [23]. However, the proportion represented by CD38 activity in the whole NADase activity was not quantified. In healthy humans, CSF and brain NAD levels are decreased with aging, which could be interpreted as increased CD38 levels [7,8].

CD38 expression was recently found to be driven by a cocktail of cytokines and chemokines secreted by senescent cells. Cellular senescence is a cell fate in which cells cease to divide and adopt a senescent-associated secretory phenotype (SASP) characterized by the secretion of factors (including proinflammatory cytokines and chemokines) that promote chronic “sterile” inflammation and fibrosis [68]. Several studies found that senescent cells accumulate with age, including in the brain [69]. Of note, brain tissues from multiple sclerosis, Alzheimer’s disease, and amyotrophic lateral sclerosis patients showed increased numbers of senescent cells, mostly astrocytes [70,71]. Moreover, senolytic treatments were found to be neuroprotective in murine models of Alzheimer’s disease [72,73]. Noteworthy, SASP factors secreted by senescent cells were found to increase CD38 expression in peripheral macrophages [74]. Assuming that microglial cells, the resident brain macrophages, react like peripheral macrophages, we can hypothesize (Figure 2) a mechanism by which accumulation of senescent astrocytes and SASP proinflammatory factors release promote increased expression of CD38 in astrocytes and microglial cells as previously described [48,52], leading to amplified proinflammatory cytokine release and NAD depletion caused by increased CD38 enzymatic activity. NAD depletion leads to reduced activity of NAD-dependent enzymes, including reduced poly (ADP-ribose) polymerase (PARP) activation and accumulation of DNA mutations, as well as reduced sirtuins activity, leading to mitochondrial dysfunctions and oxidative stress [9]. In such a scenario, acting on CD38 could stop neuroinflammation (which is in line with observations in CD38 KO mice) as well as NAD depletion, thus resulting in neuroprotection.

## 6. Conclusions

While direct evidence implicating CD38 in NDDs is still lacking, CD38 nevertheless represents a very promising target to fight neurodegeneration as well as neuroinflammation due to its close link with aging and experimental evidences from CD38 KO mice. Since CD38 controls brain NAD bioavailability and the activity of NAD-dependent enzymes that are crucial for neuronal survival, inhibition of CD38 enzymatic activity leading to increased NAD levels might be of interest to treat NDDs. Unfortunately, identified small molecule inhibitors of CD38 enzymatic activity either have an IC_50_ in the micromolar range [51,64], does not cross the BBB like the compound 78c [75], or trigger antibody-dependent cell-mediated cytotoxicity like the anti-CD38 antibodies daratumumab or isatuximab [76] (for an extensive review of pharmacological CD38 modulators, see [77]). The immunosuppressive effect of anti-CD38 antibodies like daratumumab on plasma cells and plasmablasts could be useful against autoimmune neurological disorders like multiple sclerosis [78]. Nevertheless, we still need to better understand the physiological role of CD38 in brain cells and, more particularly, what the consequence is of CD38 deletion/inhibition on brain cADPR, ADPR, NAADP, and ADPRP levels to know exactly how to modulate CD38 to maximize efficacy and lower potential adverse events (like the social behavior alterations observed in CD38 KO mice caused by reduction of cADPR in specific brain subparts).

## Figures and Tables

**Figure 1 cells-09-00471-f001:**
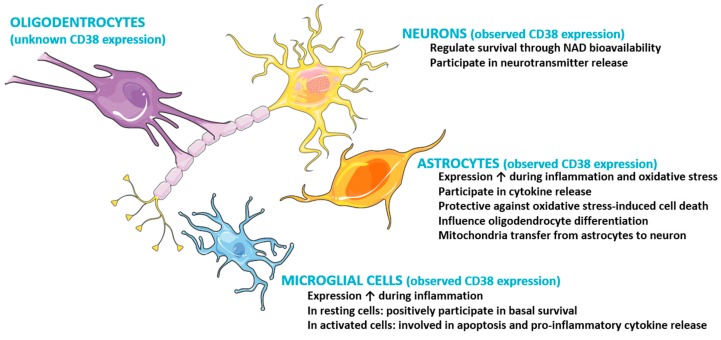
Cartoon summarizing current data on CD38 expression and functions in neuronal and glial cells.

**Figure 2 cells-09-00471-f002:**
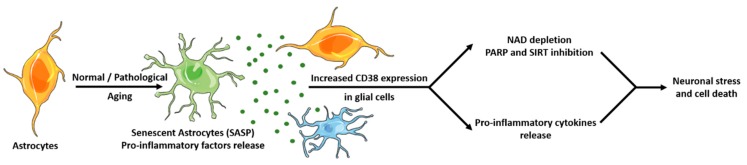
Hypothetic role played by CD38 in neurodegeneration and neuroinflammation. Normal or pathological aging is characterized by an increased number of senescent cells in the brain, mostly astrocytes, which adopt a senescence-associated secretory phenotype (SASP) characterized by the release of proinflammatory cytokines and chemokines. These proinflammatory factors increase CD38 expression in astrocytes and microglial cells, leading to (i) amplified release of proinflammatory cytokines and neuroinflammation as well as (ii) NAD depletion due to increased CD38 enzymatic activity that results in reduced activity of the NAD-dependent enzymes sirtuins and PARP, accumulation of DNA damage as well as metabolic dysfunctions and oxidative stress, leading to neuronal impairment and ultimately cell death.

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
