# Peer review of "CD38 in Neurodegeneration and Neuroinflammation"

_cells, 2020, doi:10.3390/cells9020471_

Round 1

Reviewer 1 Report

accept in current form

Author Response

We thank reviewer #1 for acepting our manuscript in current form

Reviewer 2 Report

This is a review on CD38 expression and functional activity in brain cells, particularly, in chronic neurodegeneration. It summarizes data on CD38 enzymatic activity and the role of cADPR in the regulation of neurotransmitters release as well as on CD38-mediated neuroinflammatory mechanisms. The manuscript is well-organized, but I suggest to correct some importants issues:

  1. Figure 1 itself and it's legend contain wrong notes about "SASP secretion". It seems to be the authors' misunderstanding which should corrected. SASP is a senescence-associated secretory phenotype, and it can not be secreted (!). SASP means secretion of cytokines and chemokines that are responsible for local inflammatory reactions.
  2. Physiological role of CD38 receptor in the brain is very briefly explained. The authors should detalize their view on CD31-CD38 interactions in brain neurons. Particularly, they should provide data on CD38 on the plasma membrane of neurons as well as on availability of CD31 for such kind of interactions within the neurovascular unit.
  3. I recommend to add a figure to illustrate current data on CD38 expression and functions in neuronal and glial cells.
  4. Also, some information on possible therapeutic effects of CD38 ligands or modulators should be discussed and summarized (i.e. in a table).

Author Response

We warmly thank Reviewer #2 for its comments on our manuscript. Please find below the modifications that were added to address the remarks of Reviewer #2:

  • Reviewer #2 noticed that there was a misuse of the SASP term and we warmly thank Reviewer #2 for that. Accordingly, we modified chapter 5 by adding/modifying the following sentences (underlined text): “Cellular senescence is a cell fate in which cells cease to divide and adopt a senescent-associated secretory phenotype (SASP) characterized by the secretion of factors (including pro-inflammatory cytokines and chemokines) that promote chronic “sterile” inflammation and fibrosis” (line 216 using the “Track changes” mode); “Noteworthy, SASP factors secreted by senescent cells were found to increase CD38 expression in peripheral macrophages” (line 223); “Assuming that microglial cells, the resident brain macrophages, react like peripheral macrophages, we can hypothesize (Figure 2) a mechanism by which accumulation of senescent astrocytes and SASP pro-inflammatory factors release promote increased expression of CD38 in astrocytes and microglial cells as previously described [48,53], leading to amplified pro-inflammatory cytokines release and NAD depletion caused by increased CD38 enzymatic activity” (line 224).
  • The legend of Figure 2 was also modified: “Normal or pathological aging is characterized by an increased number of senescent cells in the brain, mostly astrocytes, which adopt a senescence-associated secretory phenotype (SASP) characterized by the release of pro-inflammatory cytokines and chemokines. These pro-inflammatory factors increase CD38 expression in astrocytes and microglial cells, leading to (i) amplified release of pro-inflammatory cytokines and neuroinflammation as well as (ii) NAD depletion due to increased CD38 enzymatic activity that results in reduced activity of the NAD-dependent enzymes sirtuins and PARP, accumulation of DNA damage as well as metabolic dysfunctions and oxidative stress; leading to neuronal impairment and ultimately cell death”.
  • The text below the SASP astrocyte of Figure 2 was also modified: “Senescent Astrocytes (SASP) Pro-inflammatory factors release

  • Reviewer #2 asked us to detail our view of CD31-CD38 interaction in the brain. We modified the text accordingly by reformatting chapter 3.3: “CD31 expression in the brain is restricted to vascular endothelial cells [42] where it plays a key role in the establishment and the maintenance of intercellular junction between endothelial cells through homophilic binding, as well as regulate vascular permeability [43]. Direct interaction between CD31 on endothelial cells and CD38 on brain cells surrounding cerebral vasculature (pericytes) could have a role in the establishment and/or the maintenance of the neurovascular unit, but no data currently support such assumption.
  • CD31 can be cleaved by metalloproteinases, released as a soluble fragment (sCD31) [44], and high sCD31 levels are detected in human plasma samples [45]. Due to the presence of the blood blood-brain barrier (BBB), sCD31-CD38 interaction in brain cells seems highly unlikely. Yet, since (i) sCD31 is detected in human CSF [46] and (ii) CD38 is mainly located at the plasma membrane level in brain cells [21,33], sCD31 could interact with CD38 located at the plasma membrane of brain cells to trigger CD38 internalization. The physiological function of this putative sCD31/CD38 interaction in brain cells remains elusive. It is worth noting that in pathological conditions in which BBB breakdown is observed like after ischemic stroke, sCD31 levels strongly increased in the brain, and were found to correlate positively with (i) neurological stroke severity and (ii) the degree of functional disability [46,47]. From this standpoint, sCD31-CD38 interaction might be harmful to neurons following BBB breakdown, but clear data are lacking. Accordingly, it is interesting to note that plasma sCD31 levels are increased in several neuroinflammatory diseases including multiple sclerosis and HIV-encephalitis [18], but whether such an increase is also observed in CSF remains unknown.

  • Reviewer #2 recommended to add a figure summarizing the data on CD38 expression and functions in neuronal and glial cells. We added Figure 1 in the main text.

  • Reviewer #2 recommended to add some information on possible therapeutic effects of CD38 ligands or modulators using a Table. Since such a table was already published in a recent review entitled “The Multi-faceted Ecto-enzyme CD38: Roles in immunomodulation, cancer, aging, and metabolic diseases.” (Front. Immunol. 2019), we added the reference to this publication (line 253): “Unfortunately, identified small molecule inhibitors of CD38 enzymatic activity either have an IC50 in the micromolar range [52,65], does not cross the BBB like the compound 78c [76], or trigger antibody-dependent cell-mediated cytotoxicity like the anti-CD38 antibodies Daratumumab or Isatuximab [77] (for an extensive review of pharmacological CD38 modulators, see [78]). The immunosuppressive effect of anti-CD38 antibodies like Daratumumab on plasma cells and plasmablasts could be useful against autoimmune neurological disorders like multiple sclerosis [79].” 

This manuscript is a resubmission of an earlier submission. The following is a list of the peer review reports and author responses from that submission.

Round 1

Reviewer 1 Report

This review aims to link CD 38 and neurodegenerative diseases, through summarizing recent literature published regarding the topic. There are many paragraphs not very closely related to the topic, but rather distracting, e.g. explaining the diseases, as well as other interacting molecules, NAD, all these enzymes, any one of the topics can be a review topic itself. Tt will take away the attention of CD38 in neurodegenerative disease role. 

Reviewer 2 Report

This is a review aim at discussing the potential role of CD38 in Brain physiology and pathology. Although this review is in accordance with the literature the authors do not really provide a critical review of the literature and appear to take most of the published material by heart. In a review it is very important to truly go through the papers in the literature and analyze the dat5a and make sure that the cited references really do the job of looking at what is commented.

Just as an example in line 318 and 319 the authors comment that the study by Braidy looks at CD38 activity in the brain, the authors should actually look at the experiments and also at the methodology used in the paper. In the particular study mention what is measure is brain NADse activity and not CD38 activity. To determine CD38 activity other steps ate necessary (for example show specific activity by using internal controls such as activity of a CD38 KO animal or use of specific CD38 inhibitors). NADase activity does not mean CD38 activity unless the proper controls are demonstrated.

Also the authors should be critical of the literature in regards of cADPR and NAADP as calcium messengers and really look and analyze the data that is been presented on their review. For example did the original manuscripts use the proper controls to really demonstrate the  effects of the calcium "messengers". For example 8-BR-cADPR is not necessarily a great control inhibitor for CADPR since it can be rapidly degraded to 8-BR-ADPR that may inhibit TRP channels.

It is very important to consider that perhaps many of the effects attributed to cADPR maybe actually caused by ADPR (that is generated in much higher abundance and can activate membrane calcium channels.

Thus I believe that this is an important review but at the present form does not provide what I review should do that is : critically appraise the literature and not just accept what authors propose in their original manuscripts.

Also since this is a review the authors should make the effort to cite as much as possible original work and not other reviews.

Reviewer 3 Report

This is an excellent review about a very important and somewhat neglected aspect of neurodegenerative diseases and NAD metabolism in the brain. It strongly suggests that modulation of CD38 activity could be one mechanism of managing neurodegeneration. The section “CD38 involvement in neurodegeneration and neuroinflammation: evidence” is particularly strong and really addresses this issue. This section shows (1) that there is so much compelling evidence for a role of this enzyme in brain health and disease and addressing these pathways could in the future become a treatment option and (2) but before that, more research is urgently needed. Figure 1 is a great summary of the article.

There are some key point that need to be addressed (see below), plus minor suggestions.

Needs to be addressed:

Abstract:

One critical aspect of review is missing – the fact that NAD supplementation is beneficial in NDDs. This is one of the key points in the review and should be mentioned in the abstract.

Main text:

Line 68-70 (re: internalization of CD38): A bit more information on mechanisms on CD38 internalization is needed (clathrin coated pits, ligand-induced internalization, etc). Is it also ubiquitinated and if yes, via which ligase (E3?) It would also be good to have some kind of concluding sentence at the end of this paragraph stating that internalization of CD38 will lead to lowered NAD (even though this is explained in detail later).

Line 94: Is there any data suggesting that neuroinflammation changes trafficking of CD38? Would be good to state this here.

Line 121-122: Which factors regulate the conformational change of CD38? Please describe here.

Line 275-276… Other enzymes and pathways also regulate NAD levels, which may also be important in NDDs. Add a sentence stating this and that CD38 may be one of several mechanisms involved in the low levels of NAD in NDDs.

While this review is focussed on neurodegeneration, cancer should be mentioned as well somewhere due to the potential roles of CD38 in glioma/glioblastoma.

Conclusions:

Line 352: Inhibition of CD38 could be a treatment strategy… Are there known inhibitors of CD38? Would be great to discuss briefly if there are any. If yes, what are their medicinal chemistry properties? Can they be made to pass the BBB?

Minor points:

Main text:

Line 40-42: The following sentences are written as though they contradict each other, but they do not. “Indeed, on one side, NAD  levels were found to decrease as a consequence of aging [6], including in the human brain [7]. On the other side, NAD was found to be a potent neuroprotective and anti-inflammatory molecule [8].”

Line 107-108: Please change “having a detrimental effect on apoptosis” to “increases apoptosis” or similar (since apoptosis can sometimes be a good thing).

Line 164-165: This sentence is long and confusing: “Whether CD31-CD38 interaction might be physiologically observed in brain cells is highly unlikely since CD31 is not express in the brain (except in the lumen of brain vessels) and due to the presence of the blood brain barrier (BBB).”

Line 316: HD is different from the other diseases due to its very strong genetic cause (autosomal dominant). Perhaps talk about this in one sentence – that HD has  faster progression /earlier onset but that NAD levels may influence the progression.

Conclusions:

Line 349: I suggest to add the word genetic: “While direct GENETIC evidence….” Since there is a lot of very strong other evidence.

Round 2

Reviewer 1 Report

The reply from them did not address my concern of being focused on CD38, but rather elaborated on different topics. The scientific aspects of this review is not very high, because this paper did not focus on CD38.

Rather, given the non-disclosed commercial benefits of their founded company, they should completely honest and open about their conflict of interests. And scientifically, this review looks like a lay audience summary instead of a peer review.

Reviewer 2 Report

The manuscript is OK now.